# Effect of a Pedometer-Based, 24-Week Walking Intervention on Depression and Acculturative Stress among Migrant Women Workers

**DOI:** 10.3390/ijerph16224385

**Published:** 2019-11-09

**Authors:** Youlim Kim, Young-Me Lee, Mikyeong Cho, Hyeonkyeong Lee

**Affiliations:** 1College of Nursing, Yonsei University, 50-1, Yonsei-ro, Seodaemun-gu, Seoul 03722, Korea; goshimak@naver.com (Y.K.); zzomi324@naver.com (M.C.); 2School of Nursing, DePaul University, Chicago, IL 60640, USA; YLEE23@depaul.edu

**Keywords:** physical activity, physical intervention, health behavior, health promotion, cognitive-behavioral therapy

## Abstract

Little research has examined exercise-based interventions meant to alleviate depressive symptoms among Korean-Chinese migrant women workers living in Korea. Thus, this study evaluated the effectiveness of a 24-week walking program on reducing depressive symptoms and acculturative stress levels in this population. This quasi-experimental sequential walking program was conducted with 132 Korean-Chinese women over a period of 24 weeks. Participants were divided into either a standard treatment group (n = 69) or enhanced treatment group (n = 63). All participants were instructed to walk using a pedometer. The enhanced treatment group also received mobile text messages designed to encourage walking adherence and improve acculturation. Participants were asked to complete two structured questionnaires, the Centre for Epidemiologic Studies Depression Scale and the Acculturative Stress Scale, to evaluate the intervention programs. At the end of the program, both groups exhibited decreased depression scores, but the decrease in the enhanced treatment group was more significant both at weeks 12 and 24. Acculturative stress was also found to have decreased at weeks 12 and 24. Our findings show the walking program reduced the depressive symptoms and acculturative stress levels among the Korean-Chinese women in this study. Further studies will be needed to analyze the relationship between walking step count and mental health considering exercise intensity.

## 1. Introduction

Depression is a common mental disorder that is indicated by a depressive emotion, loss of interest or pleasure, loss of energy, sleep or appetite disorders, and decreased concentration [1]. More than 300 million people of all ages around the world suffer from depression, and women are more likely than men to have depression [2]. Furthermore, depression remains a substantial mental health problem across migrant populations. Based on data from 25 studies, the prevalence rate among migrants was found to be 15.6% [3]. Migrant populations are recognized as being at risk for adverse mental health issues due to post-migration stressors and the process of adjusting to the culture and system of their host country [4]. Nevertheless, individuals in these populations often have limited access to mental health services due to barriers such as language, lack of familiarity with mainstream services, and cultural beliefs [5]. Although migrant populations are among the most vulnerable and often need special attention from healthcare professionals, limited research has been done on the effectiveness of interventions to improve mental health outcomes for migrant workers.

Korean-Chinese (KC) migrants are the largest immigrant group in Korea, with about 680,000 residents accounting for roughly 31% of the total immigrants in 2017 [6]. Moreover, one study found that 70.4% of KC women aged 40 years or older [5] worked as housekeepers or in restaurants [7,8]. This is significant because an earlier study on acculturative stress and depression among KC migrant workers [9] found that service workers reported higher rates of depression when compared to other types of workers, and women workers had significantly higher rates of depression than male workers. Specifically, KC migrant workers with high levels of acculturative stress were more likely to experience depressive symptoms than those who had low levels of acculturative stress.

Previous research has shown that walking is beneficial in the decrease of depressive symptoms. In a 12-week walking intervention for middle-aged Iranian women [10], an increased number of steps significantly lowered the participants’ depression severity and, in another study [11], the more the number of walking steps increased, the more the depression scores decreased. In addition, acculturation has been shown to be associated with an individual’s health behaviors [12]. Acculturative stress, which is defined as the losses that arise when adapting or assimilating to a different culture with regard to beliefs, routines, and social roles [13], is also related to adherence to health behaviors. Culturally assimilated migrants have been reported to be more physically active compared to a lower acculturation group [12].

However, achieving the same positive mental health effects with walking seems to be challenging when applied to migrant populations. Migrant populations may place a different level of importance on the benefits of physical activities [14]. Their ability to perform physical activities is often limited by a lack of time, motivation, energy, and financial resources [7,15]. 

In addition, migrant populations often struggle with low socioeconomic status and isolation from mainstream channels of communication, which can create further barriers to participation in intervention programs promoting health [16]. Therefore, cognitive behavioral strategies focused on engaging members of migrant populations to maintain recommended walking routines need to be incorporated into an intervention program that is culturally adaptive and collectively created by both participants and researchers to meet the needs of migrant populations [17]. This can be done by using a community-based participatory research (CBPR) approach to create health-related interventions. CBPR promotes co-learning and capacity building among all partners and facilitates collaborative, equitable partnerships during the entire research process [18]. As such, the research presented herein is expected to empower KC migrant communities and encourage positive health outcomes, including the decrease of cardiovascular disease risks [19]. However, the influence of culturally-adaptive CBPR-based interventions on the mental health outcomes of KC migrant women workers living in Korea is currently not well understood. Thus, we aim to examine the efficacy of a culturally-adaptive 24-week home-based walking intervention on reducing acculturative stress levels and depressive symptoms among KC migrant women workers.

## 2. Materials and Methods 

This study used a quasi-experimental sequential design and participants were assigned to either the standard treatment (ST) or enhanced treatment (ET) group. A total of 132 middle-aged female KC migrant workers participated. This study was conducted from May 2012 to April 2015; ST data were collected in 2013 and ET data were collected in 2014. The study protocol is described in detail in Cho et al.’s work [17].

### 2.1. Participants

Participants were recruited through posters, word of mouth or the distribution of leaflets at three KC churches, a migrant support center, and two KC markets. The inclusion criteria used for this study selected KC migrant women workers who were between 40 and 65 years of age; had worked full-time during the previous six months; spoke Korean; and had regular access to a mobile phone.

KC women who usually engaged in regular exercise (i.e., 30 minutes a day more than three days a week during the past three months) or with medical restrictions in terms of physical activity readiness were excluded. Physical activity readiness was measured using the Physical Activity Readiness Questionnaire, which tests minimum readiness for moderate physical activity programs [18]. 

All participants in both the ST and ET groups received a pedometer to use during the 24-week walking intervention. During a 12-week adaptation period, the participants received assistance to further promote adherence to the program’s walking routine. This was followed by a 12-week maintenance period in which the participants were expected to continue program adherence on their own.

### 2.2. Walking Adherence

Walking adherence was measured twice, at 12 and 24 weeks, and was defined as the average number of steps per day over the previous 12 weeks. The daily number of steps was measured with a pedometer (DIGI-WALKER CW-700/701; Yamax, Japan). Steps less than 300 count were considered invalid data because it meant that the pedometer was either not used or had malfunctioned [20]. Each week needed to have three or more days with valid data of at least 300 steps a day to be counted [21]. 

### 2.3. Depression

The Korean version of the Center for Epidemiologic Studies Depression Scale [22,23] was used to measure the level of depression experienced by the participants. This assessment tool consists of 20 items used to measure an individual’s experience of depression during the previous week. It uses a four-point scale (nearly = 0 points, sometimes = 1 point, significantly = 2 points, and mostly = 3 points), with scores ranging from 0–60. Higher scores indicate higher levels of depression. Cronbach’s α ranged from 0.85 to 0.90 in the original study [18] and Cronbach’s α for this study sample was 0.92.

### 2.4. Acculturative Stress

Acculturative stress was measured with the Acculturative Stress Scale for International Students [24], which was translated into Korean with an established acceptable internal consistency reliability (Cronbach’s α of 0.94) by Yang [25]. It consists of 36 total items: perceived discrimination (8 items), homesickness (4 items), perceived hate/rejection (5 items), fear (4 items), culture shock (3 items), guilt (2 items), and miscellaneous (10 items). We only used 35 items, excluding one item related to discrimination based on skin color, as it was not relevant to our study. Each item was rated on a five-point scale ranging from 1 (strongly disagree) to 5 (strongly agree), and higher scores indicated higher levels of acculturative stress. Cronbach’s α was 0.92 in this study.

### 2.5. Intervention

#### 2.5.1. Standard Treatment

The intervention was developed using the Intervention Mapping method [26] to enhance participants’ social-psychological and cognitive competence, as well as to improve their cultural adaptation and exercise adherence. Program orientation and baseline assessment were conducted by trained research assistants. Participants were provided with program manuals, a pedometer, and training in walking and stretching. Based on the guideline by the American College of Sports Medicine [27], in the program orientation, the participants were instructed to walk with moderate intensity to maximize the psychological benefits of exercise. Using this guideline, we encouraged participants to walk one step every two seconds. A trained nurse was consulted when setting the daily step goal. According to the number of steps taken per day, the physical activity of the participants was classified into five levels: sedentary (<5000 steps/day); low active (5000–7499 steps/day); somewhat active (7500–9999 steps/day); active (10,000–12,499 steps/day); and highly active (≥12,500 steps/day) [28]. The established goal was to increase the number of steps taken by more than 3000 per day within four weeks [29]. This goal was implemented three times; at baseline, week 4 and week 8. Participants were asked to record their number of daily steps every day for 24 weeks and to report this information to the research team via mobile phone every 2 weeks.

#### 2.5.2. Enhanced Treatment

During the 12-week adaptation period, only the ET group received 12 exercise-related motivational text messages and 12 medal images (such as gold, silver, or bronze) to encourage exercise adherence. To improve cultural adaptation, informational illustrations portraying real-life experiences necessary to living in Korea, such as how to order drinks in cafés and how to read dry-cleaning labels, were identified by a needs survey. Those illustrations, which make cultural information easy to understand, were sent to participants’ mobile phones once during every 2-week period [17].

### 2.6. Data Collection

The study was approved by the ethics committee from the institution where the research was conducted (No. 2012-0008). Prior to the collection of data, trained research assistants explained the purpose of the study, highlighting the need for participant engagement in research, and then obtained written informed consent. The participants were informed of their right to voluntary participation and that they could stop participation at any point in the study. In order to preserve anonymity, participants’ names cannot be identified on any data collection sheets. To help ensure all participants understood the terms of the study, face-to-face data collection was conducted by trained Korean research assistants using structured questionnaires. All data (except for demographic characteristics) were measured at the beginning of the study for a baseline, then again at weeks 12 and 24.

### 2.7. Statistical Analyses

Data were analyzed using IBM SPSS Statistics 22.0. Descriptive statistics (frequency, means, and standard deviations) were used to describe participants’ basic characteristics. Group differences were analyzed using independent *t*-tests and chi-squared tests. The walking steps, depression, and acculturative stress scores of 12 weeks and 24 weeks compared to the baseline were conducted by paired *t*-test. Since missing data appeared at various points in the three measurements, a linear mixed model analysis including the missing data was used to confirm the interaction between time and group. We divided the average number of steps by 1000 at three times (baseline, 12 weeks, and 24 weeks) and used the linear mixed effect analysis as a continuous variable to analyze the effect on depression and acculturative stress. The number of steps was not analyzed according to each step, but according to 1000 steps that were combined into a unit based on previous studies [30,31]. Significance was set at 0.05 for all tests. To assess the influence of the baseline imbalance on attrition [32], a propensity score matching technique was used. We calculated the propensity scores of the “attrition group” and the “retention group” through a logistic regression analysis according to the participants’ baseline general characteristics (age, education level, income, marital status, duration of stay, chronic diseases, body mass index [BMI], etc.). 

## 3. Results

### 3.1. Participants’ Characteristics and Prospensity Scores Matching

We initially recruited 264 KC women to participate in this study; however, 80 did not meet the eligibility criteria and 52 did not wish to participate. Therefore, we were left with 132 participants in total. The recruitment and retention of participants is described in an earlier publication of this research project [19].

Participants’ general characteristics and baseline walking steps, score of depression, and acculturative stress are shown in Table 1. A homogeneity test was conducted to compare general characteristics such as age, residence time, education, and income for the ST group and the ET group. The average age of the participants was 56.20 ± 5.46 years in the ST group and 56.62 ± 4.69 years in the ET group (*p* = 0.639). The average duration of the stay in Korea for the participants was 81.41 ± 49.18 months in the ST group and 126.44 ± 77.86 months in the ET group (*p* < 0.001). The average months of working at their current job was 54.06 ± 40.40 in the ST group and 61.08 ± 57.80 in the ET group (*p* = 0.419). The average time per day the participants spent working was 11.71 ± 3.17 hours in the ST group and 13.24 ± 4.69 hours in the ET group (*p* = 0.030). The number of high school graduates was 40 (58.0%) in the ST group and 41 (65.1%) in the ET group, and there was no difference between the two groups (*p* = 0.402). The monthly income was 1312.54 ± 147.87 in the ST group and 1332.64 ± 359.31 dollars in the ET group, and there was no difference between the two groups (*p* = 0.670). Regarding participants’ types of jobs, housekeeping accounted for the highest proportion (n = 106, 80.3%), followed by restaurant waitressing (n = 13, 9.8%). Other types of jobs included caregiving, self-employment, and so on (n = 13, 9.8%). The number of participants with no chronic diseases was 63 (47.7%), and there was no difference between the groups (*p* = 0.745). 

Of all the participants, 84 out of 132 KC women completed the 24-week follow-up test (overall attrition rate = 36.4%) Therefore, propensity score matching (PSM) analysis was conducted to increase the validity of the intervention by confirming the bias of the retention group. Before matching, a total of 48 participants were in the attrition group, while 84 participants were in the retention group. The PS was calculated using general characteristics variables, and the PS of the retention group was distributed from 0.4 to 1.0, with an average score of 0.63. The PS of the attrition group was distributed between 0.3 and 0.8, with an average score of 0.61. After one-to-one matching according to the PS score, 46 dropouts were paired with 46 retainers. The PS average of the retention group was 0.62 and the PS average of the attrition group was 0.60, and the distribution was similar between the two groups.

The PS distribution of participants before and after matching and after controlling for general characteristic variables, such as age, duration of stay, working time, income, and education level by covariates, is shown in Figure 1.

### 3.2. Walking Adherence, Depression, and Acculturative Stress

Figure 2 shows the walking adherence, depression, and acculturative stress from the baseline to weeks 12 and 24 in both groups. The comparison between groups was based on the baseline, 12-week, and 24-week average scores. Compared to the baseline, the number of walking steps significantly increased in both the ST and the ET groups at week 12 (ST: t = 7.473, *p* < 0.001 ET: t = 5.649, *p* < 0.001 and at week 24 (ST: t = 7.668, *p* < 0.001 ET: t = 3.252, *p* = 0.003). The participants’ depression significantly decreased at 12 and 24 weeks compared to the baseline in the ET group (at week 12: t = −3.244, *p* = 0.002, at week 24: t = −3.368, *p* = 0.002). Similarly, participants’ acculturative stress significantly decreased at 12 and 24 weeks compared to the baseline in the ET group (at week 12: t = −2.393, *p* = 0.021, at week 24: t = −2.464, *p* = 0.018).

### 3.3. Changes in Depression and Acculturative Stress

Table 2 reflects the linear mixed effects models for the depression and acculturative stress outcomes after adjusting for baseline characteristics of age, monthly income (USD), type of job, duration of stay, education, working time per day, and chronic disease. In the linear mixed model analysis, the number of walking steps—measured three times—was included, and as the daily steps increased by 1000 steps, all participants showed a significant reduction in depressive symptoms and acculturative stress. In addition, the analysis showed a significant interaction effect of group and time for depression at weeks 12 and 24, compared to the baseline. Similarly, there was a significant effect of walking on acculturative stress reduction for all participants. A significant interaction effect between group and time for acculturative stress was shown at weeks 12 and 24 compared to the baseline.

## 4. Discussion

This study was designed to evaluate the effect of a pedometer-based 24-week walking program on depression and acculturative stress among KC migrant women workers. While both groups had significantly increased their walking steps at 12 or 24 weeks compared to the baseline, depression and acculturative stress decreased in the ET at weeks 12 and 24 due to significant interactions between time and group. This suggests that the intervention enhanced by socio-cognitive psychological factors that was applied to the ET was effective.

This study demonstrated the effectiveness of a home-based walking intervention with the use of a pedometer to enhance cultural adaptation and decrease the levels of depression and acculturative stress experienced by middle-aged women KC migrant workers. Given the continuous increase in the number of KC migrant workers coming to Korea, the need to better understand how exercise interventions improve their mental health is important and timely. This study is among the first to examine the effects of a walking-based intervention, which has been shown to be a protective factor against depression [33], on commonly experienced symptoms, such as depression and acculturative stress, among KC migrant workers living in Korea. As such, these findings make a positive contribution to the existing body of literature.

Compared to the baseline, the walking-based intervention showed a positive effect on the reduction of depressive symptoms at 24 weeks, which is in line with findings for similar interventions [21,33,34,35]. Notably, we observed a sustained reduction in depressive symptoms for the ET group only. These results suggest that participants who have an additional cultural adaptation intervention paired with walking regularly experience fewer depressive symptoms and show less acculturative stress than those who only walk. 

In our study, the number of steps a day was measured using 1000 steps as a unit. Gilson et al. [30] as well as Butler et al. [31] encouraged participants to walk an additional 1000 steps daily to promote physical activity in the workplace. Walking an additional 1000 steps per day is also associated with a lower BMI [36] and improvement in self-reported depressive symptoms [37]. Likewise, we found that when the number of steps was increased by 1000 per day, participants’ depressive symptoms decreased. Therefore, we recommend a progressive increase of at least 1000 steps per day for people who walk fewer than 10,000 steps in order to promote the health benefits associated with walking.

Concerning acculturative stress, the significant reduction in the earlier phase of the study was maintained through to week 24. KC migrant women experience a variety of adaptation challenges and stressful life events in the workplace and at home. It has been reported that members of migrant populations commonly encounter difficulties adjusting to numerous changes in their lives [38]. The high levels of acculturative stress that accompany this process of adaptation may result in increased mental health concerns. Therefore, in addition to the provision of walking intervention, a supportive environment at work and in the community is needed to help migrant populations adapt to their new culture.

CBPR has been used as an effective way of approaching health interventions in migrant populations [39,40]. As such, the CBPR approach was adopted by this study particularly to address the challenges of delivering culturally-appropriate walking interventions. For example, the study participants were asked to name the walking program and prioritize topics that would be most applicable to KC women. This process was appropriate to allow them to have ownership of the intervention as well as the commitment to complete it. However, we still had a considerable attrition rate of 36.4%, although attrition in some form is a nearly universal reality in longitudinal research with human participants. In this study, no different characteristics were identified between the retention group and attrition group; however, strategies to decease the attrition rate should be considered in future studies, as high attrition can threaten the external validity of an intervention by producing a final sample that is not representative of the population sample [41].

This study included several limitations. Primarily, it was difficult to design a randomized controlled trial (RCT) study in a KC migrant community. As mentioned in an earlier related study [19], it is possible that the participants in the intervention group sometimes interact with participants in the control group, which creates bias; however, this is nearly impossible to avoid due to social interactions among individuals in the same residential area [42]. In this study, the number of steps also increased significantly in the ST group provided with the same pedometer as the ET group. According to the systematic literature review of Bravata et al. [43], the use of pedometers tends to increase a person’s motivation for walking. Similarly, in a study of low socioeconomic groups [44], physical activity level was found to increase significantly in the control group, which only received basic health services. Therefore, it is assumed that providing basic intervention to individuals with low awareness of health care services has motivated them to adhere to health promotive behavior. In addition, since the participants are engaged in jobs that require high occupational physical activity—such as housekeeping and waitressing—one should be careful in extrapolating the findings to those who require low occupational physical activity, such as office workers. Since this was an intervention study of home-based walking within a community setting, the influence of exogenous variables may exist due to limited control over participants. Also, recruitment and retention may also have been limited because KC migrant workers were difficult to reach out to due to a limited budget and time schedule. Migrant populations are often considered hard-to-reach groups, and more effective strategies may be needed to overcome these difficulties. Such strategies may include using pre-existing community networks and building trust within the migrant population [16]. Further research is needed to determine practical solutions to better engage the KC migrant community.

Migration is not a cause of mental health issues in and of itself, but it may lead to individuals being exposed to many stressors associated with settling into a new host country. These circumstances contribute to an increase in acculturative stress levels. Those increased levels are, in turn, related to the increase of depressive symptoms among members of migrant populations. Thus, nurses and community leaders need to be aware of these mental health issues and work to prevent and manage the adverse effects associated with migration. With migration increasing globally, health professionals working with migrant populations should offer regular walking (based on a cognitive behavioral approach) as a way to promote positive mental health.

## 5. Conclusions

This study demonstrated that a 24-week walking intervention was suitable for decreasing depressive symptoms and levels of acculturative stress in middle-aged KC migrant women workers who were not currently prioritizing their health. Culturally adaptive triggers to motivate the KC workers to continue exercising were effective, but they may need to be updated further depending on the participants’ needs. Depression is a commonly experienced mental health concern among migrant populations worldwide, and the walking intervention presented in this study—which was further enhanced by social-cognitive elements—may be useful for groups of other ethnicities as well. In this study, walking measurements relied on participants’ self-reporting of the number of steps. Further studies are needed to examine the influence of moderate intensity walking on mental health using objective measures of walking. 

## Figures and Tables

**Figure 1 ijerph-16-04385-f001:**
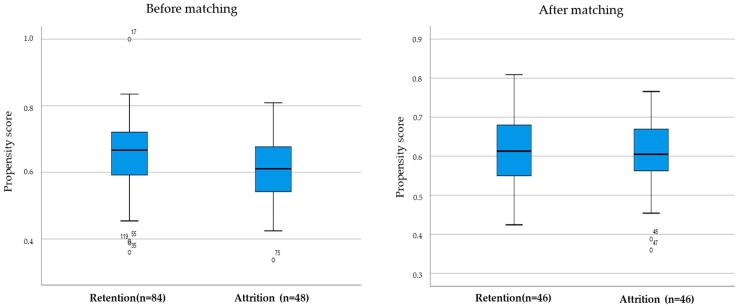
Comparison of propensity score retention and attrition.

**Figure 2 ijerph-16-04385-f002:**
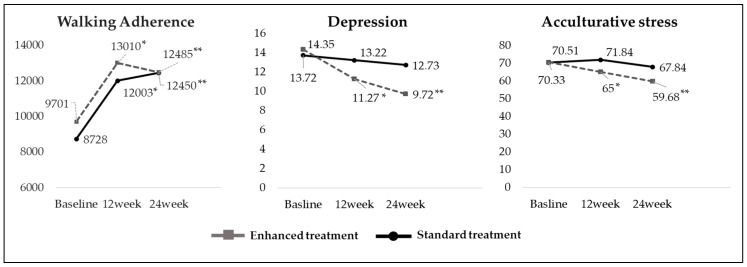
Comparison of walking adherence, depression, and acculturative stress levels between enhanced treatment (ET) and standard treatment (ST) groups over the 24-week study. * *p* < 0.05: difference between baseline and 12week, ** *p* < 0.05: difference between baseline and 24week.

**Table 1 ijerph-16-04385-t001:** Participants’ baseline characteristics.

Variable	Total	ST Group (n = 69)	ET Group (n = 63)			
	n (%)/M(SD)	n (%)/ M(SD)	n (%)/ M(SD)	*x^2^*	t	*p*
Age	54.60 (5.09)	56.20 (5.46)	56.62 (4.69)		0.47	0.639
Duration of stay (months)	102.90 (68.08)	81.41 (49.18)	126.44 (77.86)		4.01	**<0.001**
Duration of current job (months)	57.44 (49.46)	54.06 (40.40)	61.08 (57.80)		0.81	0.419
Working time(h/day)	12.45 (4.03)	11.71 (3.17)	13.24 (4.69)		2.20	**0.030**
Income (USD/month)	1322.13 (269.36)	1312.54 (147.87)	1332.64 (359.31)		0.43	0.670
Education						
≥High school	81 (61.8)	40 (58.0)	41 (65.1)	0.702		0.402
Type of job						
Housekeeper	106 (80.3)	58 (84.1)	48 (76.2)	1.443		0.486
Waitress	13 (9.8)	5 (7.2)	8 (12.7)			
Others	13 (9.8)	6 (8.7)	7 (11.1)			
Chronic disease						
None	63 (47.7)	32 (46.4)	31 (49.2)	0.106		0.745
Walking(steps/day)	9,193 (3212)	8,728 (2,977)	9,701 (3,875)		1.65	0.102
Depression	13.02 ( 8.68 )	13.72 (7.12)	14.35 (10.17)		0.41	0.681
Acculturative stress	70.42 (21.37)	70.33 (19.10)	70.51 (23.77)		0.05	0.963

Bold indicates statistically significant *p*- values (*p* < 0.05).

**Table 2 ijerph-16-04385-t002:** Changes in depression and acculturative stress scores over time.

Outcome	B	SE	t	*p*
**Depression**				
Steps/day (per 1000 steps)	−0.308	0.121	−2.545	**0.012**
ET (reference: ST)	0.701	1.551	0.452	0.652
Week 12 (reference: Baseline)	0.493	1.087	0.453	0.651
Week 24 (reference: Baseline)	−0.124	1.041	−0.119	0.906
ET*Week 12	−3.149	1.444	−2.180	**0.031**
ET*Week 24	−3.104	1.380	−2.249	**0.027**
**Acculturative stress**				
Steps/day (per 1000 steps)	−0.748	0.330	−2.266	**0.024**
ET (reference: ST)	−1.255	3.894	−0.322	0.748
Week 12 (reference: Baseline)	4.119	3.132	1.315	0.190
Week 24 (reference: Baseline)	1.661	2.728	0.609	0.543
ET*Week 12	−8.679	4.202	−2.066	**0.040**
ET*Week 24	−7.178	3.612	−1.987	**0.049**

The model is adjusted for the baseline characteristics of age, monthly income, type of job, duration of stay, education, working time per day, and chronic disease (*p* < 0.05); Bold indicates statistically significant *p*-values (*p* < 0.05); * indicates interaction of Group and Time.

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
