# Peer review of "Effect of a Pedometer-Based, 24-Week Walking Intervention on Depression and Acculturative Stress among Migrant Women Workers"

_ijerph, 2019, doi:10.3390/ijerph16224385_

Round 1

Reviewer 1 Report

Dear Authors,

Thank you for the interesting and well designed study.

In the introduction section you state that "exercise is effective and beneficial in the treatment of depression". Walking is attributed to physical activity of low intensity, whereas evidence mostly focus on aerobic moderate to vigorous intensity to demonstrate some beneficial effects on mental health. Please provide some additional background that walking and counting steps cal alleviate depression and/or stress.

Please describe exclusion criteria as if you state that some participants were not eligible.

Please justify the use of the questionnaire of Acculturative stress for middle-aged migrant women workers as you state that it was translated and adopted for the international students. Reference no. 21 can not be accessed, please check if it is correct.

Please describe what was the standard treatment (medication, cognitive therapy, etc.?). Was walking and counting steps added to the standard treatment (group ET) or was it used alone as an enhanced treatment strategy and compared to standard? 

In the section "Statistical analysis" you state that two-sample t test was used. T test always compares only 2 samples. Please clarify whether it was for independent or related samples.

Please clarify, what statistical method was used in Figure 1 to compare changes of walking adherence, depression and stress? In the result section you demonstrate t test which requires pairwise comparisons (gr. 1 vs gr. 2, gr. 1 vs. gr. 3, etc). Please reconsider Repeated measures (GLM 4) instead, as it allows comparisons in time with covariates, and calculations of within (changes in time) and between (differences between 2 treatment groups in each point of time) subject effects.

In the Statistical analysis section you describe propensity score, logistic regression analysis and some "attrition" and "retention" groups not mentioned in the further analyses, whereas your results end with Table 2, mixed linear models. Please check this situation.

Please clarify abbreviations ET and ST together with Figure 1 and Table 1. Please avoid abbreviations use in the Figures title.

Section "Acknowledgements" basically provide funding sources, so my suggestion is to rename it into "Funding".

Waiting for the improved version of your paper.

Author Response

Please see the attachment. The yellow highlighting in the attached file is what we described in the manuscript.

Reviewer’ s Comments

Author’s Response

In the introduction section you state that "exercise is effective and beneficial in the treatment of depression". Walking is attributed to physical activity of low intensity, whereas evidence mostly focus on aerobic moderate to vigorous intensity to demonstrate some beneficial effects on mental health. Please provide some additional background that walking and counting steps cal alleviate depression and/or stress.

We further explained the studies on the correlation between increased walking steps and depression and stress.

Please describe exclusion criteria as if you state that some participants were not eligible.

We excluded women who usually engage in regular exercise (i.e., 30 minutes a day, three or more days a week during the past three months) or with restrictions in terms of physical activity readiness.

Please justify the use of the questionnaire of Acculturative stress for middle-aged migrant women workers as you state that it was translated and adopted for the international students. Reference no. 21 can not be accessed, please check if it is correct.

The questionnaire of Yang et al(2007), in a study of Yang et al(2009) on married migrant women, Cronbach's alpha was .936.   In a study of Lee et al(2010) on Korean-Chinese migrant workers, Cronbach's alpha value was found to be .95, and in this study was .91.

Please describe what was the standard treatment (medication, cognitive therapy, etc.?). Was walking and counting steps added to the standard treatment (group ET) or was it used alone as an enhanced treatment strategy and compared to standard?

We separately explained standard treatment.

In the section "Statistical analysis" you state that two-sample t test was used. T test always compares only 2 samples. Please clarify whether it was for independent or related samples.

Group differences were analyzed using independent t-tests and chi-squared tests. The walking steps, depression, and acculturative stress scores at 12 weeks and 24 weeks were compared to the baseline using a paired t-test.

Please clarify, what statistical method was used in Figure 1 to compare changes of walking adherence, depression and stress? In the result section you demonstrate t test which requires pairwise comparisons (gr. 1 vs gr. 2, gr. 1 vs. gr. 3, etc).

The statistical method of Figure 1 was supplemented in “2.7. Statistical analyses” area.

The walking steps, depression, and acculturative stress scores at 12 weeks and 24 weeks were compared to the baseline using a paired t-test.

Please reconsider Repeated measures (GLM 4) instead, as it allows comparisons in time with covariates, and calculations of within (changes in time) and between (differences between 2 treatment groups in each point of time) subject effects.

Since there were missing data across the measurements that were repeated three times, a linear mixed model analysis including the missing data was used to confirm the interactions between time and group.

In the Statistical analysis section you describe propensity score, logistic regression analysis and some "attrition" and "retention" groups not mentioned in the further analyses, whereas your results end with Table 2, mixed linear models. Please check this situation.

Since the dropout rate was high in this study, the difference of general characteristics of attrition group and retention group was analyzed by PS matching to confirm the validity of the results. However, the general characteristics of the attrition group and the retention group were no significant difference. We supplemented the additional explanations and figure for the PSM results.

Please clarify abbreviations ET and ST together with Figure 1 and Table 1. Please avoid abbreviations use in the Figures title.

We clarified abbreviations as follows;

-ET: Enhanced treatment.

-ST: Standard treatment

Section "Acknowledgements" basically provide funding sources, so my suggestion is to rename it into "Funding".

We renamed to “Funding”.

We noted Professor Chang-Gi Park and Professor Ki Jun Song who advised on statistical analysis in the Acknowledgements section.

Reviewer 2 Report

MS ID: ijerph-610308

Title: Effect of a Pedometer-based, 24-week Walking Intervention on Depression and Acculturative Stress among Migrant Women Workers

This manuscript examined the effect of walking intervention on depressive symptoms and acculturative stress in Korean immigrant women in middle-age adulthood. Psychological distress was assessed by self-report and walking intervention was evaluated over 24 weeks, using a pedometer. The authors observed that brisk walking could reduce the negative mental health effects of working time among the immigrant women. Below are general and specific comments about the study.

Abstract

The conclusion does not provide detailed findings. It should be constructed using data from the findings section.

Introduction

Page 2, lines 50-58.

There is a need to provide a more in-depth explanation that should address the association of objectively measured physical activity intensity levels (i.e., light, moderate, and moderate-to-vigorous physical activity and sedentary time) with depression and acculturative stress in women and some possible explanations for their mechanisms. The purpose of this study is not designed specifically for the finding of results. For example, what is the differences in the psychological distress scores between ST and ET groups or whether different physical activity intensity levels and sedentary time are independently associated with the psychological distress among the immigrant women?

Materials and Method

Page 2, line 73.

Describe additional detail about the intervention in standard treatment and enhanced treatment. This is a short follow-up study. Clarify summative and formative assessments of objectively measured physical activity and psychological distress questionnaire from 2012 to 2015. Line 78. Comma between ‘health and of’ should be omitted.

Study protocol

Page 3, lines 94-100.

The assessment of walking adherence is vague. Do the subjects attach the pedometer to their waistband or belt in the same position for 24 consecutive weeks and maintain a pedometer log, including weekends? Do the subjects remove the pedometer only while sleeping, bathing or swimming? Do aerobic steps or calories can be counted using the DIGI-WALKER pedometer? Do the subjects wear the pedometer to commute to and from work? It is important that total steps should be used as both continuous and categorical variables in the analyses. Clarify whether the psychological distress questionnaire is measured at baseline or at 24 weeks. Description of sociodemographic variables is missing. These variables should be used in adjusting for important covariates in the analysis. Lines 128-132. The sentences should be modified. I want to know what the criteria levels of daily steps should be used to estimate the increased 3,000 steps per day.

Page 4, lines 148-150. The authors mentioned that ‘all data were measured three times except for demographic variables.’ If so, the authors should undertake the analyses to adjusted for potential preexisting depressive symptoms and acculturative stress respectively. This may affect the primary outcome and bias results.

Statistical analysis

Line 156. The sentence of ‘by the increase of 1,000 steps/day’ is unclear. The authors mentioned above to increase 3,000 steps. The logistic regression analysis might be also used for analyzing the associations of levels of total steps (e.g., high vs. low or sedentary) with psychological distress scores.

Results

Lines 164-167. I would expect to see the results of attrition analyses, because the dropout rate for subjects is quite high (50%). Table 1 should be modified. For instance, whether the working time is calculated by daily or a week on average? Monthly income should be converted into US$ or euro €. Education should be combined elementary and middle school, and then divided into low and high groups. Type of jobs should be clarified exactly such as domestic worker (e.g., cleaning / caring / others), restaurant worker (e.g., waitress / chief / others), and so on. Walking, depression and acculturative stress should present their values at three-time measurements. Lines 183-193. A description of Figure 1 should be modified. Are there any differences in walking adherence between ST and ET groups at baseline and 12 weeks? The sentence ‘Compared to the baseline, the mean depression scores of the ET groups…’ is confuse. The sentence of the paragraph should be modified. Clarify the differences in psychological distress at each group on three points or between ST and ET groups on each measurement. Figure 1 should be modified. ST and ET groups in walking adherence are bold lines, but not in depression and acculturative stress. Figure 1 shows a significant difference in depression between baseline and 24 weeks in the ET group, but the authors do not mention in the text. Also, there is a significant difference in acculturative stress between baseline and 12 weeks in the ET group, but in the text the authors mentioned that it was no significant difference. Table 2 should be modified. Clarify more details for steps/day (per 1,000 steps). In the interaction effect of table 2, (reference: ST*Baseline) should be deleted because it is incompatible with the results in the ET group. I would expect to see the description of crude analysis. It is important to develop an understanding of sociodemographic factors mediating effect on psychological distress for KC immigrant women. Age, month income and type of job are missing in adjusted analysis. I would expect the authors to analyze the relationships of levels of physical activity with psychological distress at three times in an extra table. It is hypothesized that women with higher physical activity would be less likely to have depression and acculturative stress than those with low physical activity or sedentariness.

Discussion

The main findings of the study should be presented in the first paragraph of the discussion. The discussion section should be considered to the issues relating to the methods and statistical analysis. Some clarifications (e.g. level of physical activity) are required in the results section and the authors need to qualify some of their comments based on the weight of evidence provided by their statistical findings. The sample in the study is somewhat skewed because most of them are more likely to have a low socioeconomic position and do work overtime. The step values reported in the study may be higher than those with the higher socioeconomic position. These should be mentioned in the limitation. The subjects who wear pedometers may have taken somewhat more steps than normal despite being encouraged to maintain normal habits. These should be mentioned in the limitation.

Author Response

Please see the attachment. The yellow highlighting in the attached file is what we described in the manuscript.

Reviewer ’s Comments

Author’s Response

Abstract

The conclusion does not provide detailed findings. It should be constructed using data from the findings section.

We revised the conclusions of abstract.

Our findings show that a walking program reduced Korean-Chinese women’s depressive symptoms and acculturative stress levels. Further studies will be needed to examine the influence of walking on mental health considering exercise intensity.

Introduction

Page 2, lines 50-58.

There is a need to provide a more in-depth explanation that should address the association of objectively measured physical activity intensity levels (i.e., light, moderate, and moderate-to-vigorous physical activity and sedentary time) with depression and acculturative stress in women and some possible explanations for their mechanisms.

We revised the text with literature that studied the relationship between walking steps and depression and acculturation.

The purpose of this study is not designed specifically for the finding of results. For example, what is the differences in the psychological distress scores between ST and ET groups or whether different physical activity intensity levels and sedentary time are independently associated with the psychological distress among the immigrant women?

This study does not confirm the relationship between intensity level of physical activity and depression and acculturative stress.

We examined to compare the changes of depression and acculturative stress in control and experimental groups as a result of culturally-adaptive 24week home-based walking intervention.

Materials and Method

Page 2, line 73.

Describe additional detail about the intervention in standard treatment and enhanced treatment. This is a short follow-up study. Clarify summative and formative assessments of objectively measured physical activity and psychological distress questionnaire from 2012 to 2015.

Line 78. Comma between ‘health and of’ should be omitted.

We separately explained standard treatment and described as follows.

This study was conducted from May 2012 to April 2015; ST data were collected in 2013 and ET data were collected in 2014. Both walking steps and depression were measured at baseline, 12week and 24week.

We deleted “,”.

Study protocol

Page 3, lines 94-100.

The assessment of walking adherence is vague. Do the subjects attach the pedometer to their waistband or belt in the same position for 24 consecutive weeks and maintain a pedometer log, including weekends? Do the subjects remove the pedometer only while sleeping, bathing or swimming? Do aerobic steps or calories can be counted using the DIGI-WALKER pedometer? Do the subjects wear the pedometer to commute to and from work?

Participants were instructed to keep a pedometer out of water and remove it when sleeping by program manual.

In this study, all participants were guided to walk with moderate intensity, but the exercise intensity was not measured, only the walking steps were collected through a pedometer.

We described in discussion that control by exogenous variables was limited because it was a community-based intervention program.

It is important that total steps should be used as both continuous and categorical variables in the analyses. Clarify whether the psychological distress questionnaire is measured at baseline or at 24 weeks. Description of sociodemographic variables is missing. These variables should be used in adjusting for important covariates in the analysis.

We divided the number of steps by 1,000 and used the linear mixed effect analysis as a continuous variable.

Depression and acculturative stress were measured a total of three times at baseline, 12weeks and 24 weeks through questionnaires. This is described in the data collection section.

The measured depression and acculturative stress at the baseline were homogenized in the experimental group and the control group, and the general characteristic variables with significant differences between the groups were adjusted when linear mixed effect analysis.

Lines 128-132. The sentences should be modified. I want to know what the criteria levels of daily steps should be used to estimate the increased 3,000 steps per day.

We have omitted references citation.

The study related to the 3000 steps per day increase is as follows.

Tudor-Locke C, Schuna JM Jr. Steps to preventing type 2 diabetes: exercise, walk more, or sit less? Front Endocrinol. 2012;3:142.

We have cited a reference.

Page 4, lines 148-150. The authors mentioned that ‘all data were measured three times except for demographic variables.’ If so, the authors should undertake the analyses to adjusted for potential preexisting depressive symptoms and acculturative stress respectively. This may affect the primary outcome and bias results.

We adjusted for potential confounders that showed differences in the groups in LMM analysis. Based on your comments, we have further included and re-analyzed general characteristics such as age, income, and type of job. These analyses were conducted with the consulting of statisticians.

Statistical analysis

Line 156. The sentence of ‘by the increase of 1,000 steps/day’ is unclear. The authors mentioned above to increase 3,000 steps. The logistic regression analysis might be also used for analyzing the associations of levels of total steps (e.g., high vs. low or sedentary) with psychological distress scores.

We have advised participants to increase the number of steps by 3,000 to the next goal setting, but not all participants have implemented their goal steps.

Therefore, the effect of mental health on step count was analyzed in 1,000 steps per day based on previous studies (Gilson et al., 2009;Butler et al,,2015) not on the 3,000 steps per day.

In this study, we guided moderate intensity walking during orientation, but collected only simple steps count and did not measure exercise intensity with a pedometer.

Therefore, in the statistical analysis, the number of walking steps was set as a continuous variable and the linear mixed model was conducted.

Results

Lines 164-167. I would expect to see the results of attrition analyses, because the dropout rate for subjects is quite high (50%).

We supplemented the additional explanations and figure for the PSM results.

Table 1 should be modified.

Walking, depression and acculturative stress should present their values at three-time measurements.

Participants’ general characteristics and baseline walking steps, score of depression and acculturative stress are shown in Table 1.

Table 1 shows a comparison of the baseline data for the experimental and control groups. Walking, depression and acculturative stress at three-time measurements were shown in the figure 2.  

Lines 183-193. A description of Figure 1 should be modified. Are there any differences in walking adherence between ST and ET groups at baseline and 12 weeks? The sentence ‘Compared to the baseline, the mean depression scores of the ET groups…’ is confuse. The sentence of the paragraph should be modified. Clarify the differences in psychological distress at each group on three points or between ST and ET groups on each measurement.

Figure 1 should be modified. ST and ET groups in walking adherence are bold lines, but not in depression and acculturative stress. Figure 1 shows a significant difference in depression between baseline and 24 weeks in the ET group, but the authors do not mention in the text. Also, there is a significant difference in acculturative stress between baseline and 12 weeks in the ET group, but in the text the authors mentioned that it was no significant difference.

Figure 2 shows changes of walking adherence, depression, and acculturative stress from the baseline to weeks 12 and 24 in both groups. We revised the paragraph as follows.

We re-analyzed and revised of Figure 2(before Figure 1). The text has been revised as follows.

Compared to baseline, the number of walking steps significantly increased in both the ST and the ET groups at week 12 (ST: t = 7.473, p < .000, ET: t = 5.649, p < .000) and at week 24 (ST: t = 7.668, p < .000, ET: t = 3.252, p = .003). The participants’ depression significantly decreased at 12 and 24 weeks compared to the baseline in the ET group (at week 12: t = -3.244, p = .002, at week 24: t = -3.368, p = .002). Similarly, participants’ acculturative stress significantly decreased at 12 and 24 weeks compared to the baseline in the ET group (at week 12: t = -2.393, p = .021, at week 24: t = -2.464, p = .018).

Table 2 should be modified. Clarify more details for steps/day (per 1,000 steps). In the interaction effect of table 2, (reference: ST*Baseline) should be deleted because it is incompatible with the results in the ET group. I would expect to see the description of crude analysis. It is important to develop an understanding of sociodemographic factors mediating effect on psychological distress for KC immigrant women. Age, month income and type of job are missing in adjusted analysis. I would expect the authors to analyze the relationships of levels of physical activity with psychological distress at three times in an extra table. It is hypothesized that women with higher physical activity would be less likely to have depression and acculturative stress than those with low physical activity or sedentariness.

We deleted “reference: ST*Baseline” in Table 2.

Based on your comments, we have re-analyzed adjusting general characteristics such baseline of age, monthly income, type of job, duration of stay, education, working time per day and chronic disease. These analyses were conducted with the consulting of statisticians.

The reanalysis results are as follows:

In the linear mixed model analysis, the number of walking steps, measured three times, was included, and as the daily steps increased by 1,000 steps, all participants showed a significant reduction in depressive symptoms and acculturative stress. In addition, the analysis showed a significant interaction effect of group and time for depression at weeks 12 and 24, compared to the baseline. Similarly, there was a significant effect of walking on acculturative stress reduction for all participants. A significant interaction effect between group and time for acculturative stress was shown at weeks 12 and 24 compared to the baseline.

Based on your comments below, we conducted an analysis after dividing level of education into high school graduate and converting income to US dollars.

For instance, whether the working time is calculated by daily or a week on average? Monthly income should be converted into US$ or euro €. Education should be combined elementary and middle school, and then divided into low and high groups. Type of jobs should be clarified exactly such as domestic worker (e.g., cleaning / caring / others), restaurant worker (e.g., waitress / chief / others), and so on.

According the reviewer’s comment, we revised the variables of working time, income, education and type of jobs.

In the survey, participants were asked to fill out their own occupation, housekeeper accounted for the highest proportion, followed by restaurant waitress. Other types of jobs included caregivers, self-employment, and so on.

We have classified the occupational groups into three groups with high proportions, and the detailed occupations are described further in the manuscript.

Discussion

The main findings of the study should be presented in the first paragraph of the discussion.

The discussion section should be considered to the issues relating to the methods and statistical analysis. Some clarifications (e.g. level of physical activity) are required in the results section and the authors need to qualify some of their comments based on the weight of evidence provided by their statistical findings.

We explained main findings in the first paragraph of the discussion as follows;

This study was designed to evaluate the effect of a pedometer-based 24-week walking program on depression and acculturative stress among KC migrant women workers. While both groups had significantly increased their walking steps at 12 or 24 weeks compared to the baseline, depression and acculturative stress decreased in the ET at weeks 12 and 24 due to significant interactions between time and group. This suggests that the intervention enhanced by socio-cognitive psychological factors that was applied to the ET was effective.

The sample in the study is somewhat skewed because most of them are more likely to have a low socioeconomic position and do work overtime. The step values reported in the study may be higher than those with the higher socioeconomic position. These should be mentioned in the limitation. The subjects who wear pedometers may have taken somewhat more steps than normal despite being encouraged to maintain normal habits.

These should be mentioned in the limitation.

Wearing a pedometer itself may lead KC women in the control group who were not given interventions to promote exercise. According to the systematic literature review of Bravata [44], the use of pedometers itself tends to lead a person’s motivation for walking . In the study of low socioeconomic groups [45], the physical activity level was also significantly increased in the control group who received only basic health services. Therefore, it is assumed that providing basic intervention to individuals with low awareness of health care services has motivated them to adhere health promotive behavior. In addition, since the participants are engaged in jobs that require high occupational physical activity   such as housekeepers and waitresses, it should be careful in interpreting the findings to those who require low occupational physical activity such as office workers.

Reviewer 3 Report

The authors should reffered (in part Introduction) to the latest data on mental health, including depression data, published by WHO.
Lack of defining concepts such as depression and acculturative stress.

Author Response

Please see the attachment. The yellow highlighting in the attached file is what we described in the manuscript.

Review’s Comments

Author’s Response

The authors should refered (in part Introduction) to the latest data on mental health, including depression data, published by WHO.

We cited the latest data on depression from the WHO fact sheet homepage.

Lack of defining concepts such as depression and acculturative stress.

We further described the definitions of depression and acculturative stress in background.

Round 2

Reviewer 1 Report

Dear Authors,

Thank you for the improved version of your manuscript. Now it is easier to follow.

In the Methods section please clarify whether you used randomization when attributing participants into ST or ET groups.

Wish you all the best in your further research!

Author Response

Review’s Comments

Author’s Response

In the Methods section please clarify whether you used randomization when attributing participants into ST or ET groups.

As we addressed that this study employed a quasi-experimental sequential design, we conveniently assigned the participants to either the ST or ET group indicating non-randomized of participants.

Reviewer 2 Report

Authors have improved their MS. Well done! 

Author Response

Review’s Comments

Author’s Response

Authors have improved their MS. Well done!

Thanks to your comments.